# NINJA trial: should the nail plate be replaced or discarded after nail bed repair in children? Protocol for a multicentre randomised control trial

Abhilash Jain,[1,2] Amy Jones,[2] Matthew D Gardiner [ID],[2,3] Cushla Cooper,[2] Adam Sierakowski,[4] Melina Dritsaki [ID],[2] May Ee Png,[2] Jamie R Stokes [ID],[2] Beverly Shirkey,[2] Jonathan Cook [ID],[2] David Beard,[2] Aina V H Greig[5]

For numbered affiliations see end of article.

**Correspondence to**
Matthew D Gardiner;
matthew.gardiner@kennedy.ox.ac.uk

## ABSTRACT

**Introduction** Trauma to the nail bed is the most common surgically treated paediatric hand injury. The majority of surgeons replace the nail plate after repairing the nail bed despite a lack of evidence to do so. Replacing the nail plate may be associated with increased postoperative infection. We will investigate the impact of replacing or discarding the nail plate on infection, cosmetic appearance, pain and subsequent healthcare use. The Nail bed INJury Analysis trial (NINJA) aims to answer the question of whether the nail plate should be replaced or discarded after surgical nail bed repair in children.

**Methods and analysis** A two-arm parallel group open multicentre randomised control trial of replacing the nail plate or not, as part of a nail bed repair, will be undertaken in children presenting within 48 hours of a nail bed injury requiring surgical repair. The coprimary outcomes are: cosmetic appearance summary score at a minimum of 4 months and surgical site infection at around 7 days following surgery. Secondary outcomes are EuroQol EQ-5D-(Y); the pain intensity experienced at first dressing change; child/parent satisfaction with nail healing and healthcare resource use. We will recruit a minimum of 416 patients (208 in each group) over 3 years. Children and their parents/carers will be reviewed in clinic around 7 days after their operation and will be assessed for surgical site infection or other problems. The children, or depending on age, their parents/carers, will also be asked to complete a questionnaire and send in photos of their fingernail at a minimum of 4 months postsurgery to assess cosmetic appearance.

**Ethics and dissemination** The South Central Research Ethics Committee approved this study on 4 June 2019 (18/SC/0024). A manuscript to a peer-reviewed journal will be submitted on completion of the trial as per National Institute for Health Research publication policy. The results of this trial will substantially inform clinical practice and provide evidence on whether the practice of replacing the nail plate should continue at the time of nail bed repair.

**Trial registration number** ISRCTN44551796.

## INTRODUCTION

Nail bed trauma is the most common surgically treated paediatric hand injury and

### Strengths and limitations of this study

► Pragmatic study design to ensure generalisability.
► First randomised trial to use the Reconstructive Surgery Trials Network.
► A health economic evaluation as well as the clinical assessment will be performed.
► It will not be possible to get patient-reported outcomes from all participants owing to their young age.

accounts for 10 000 operations annually in the UK.[1] Surgery involves removing the nail plate (fingernail) and repairing the underlying nail bed laceration with sutures. Once the nail bed has been repaired, 96% of surgeons in the UK replace the nail plate.[1] The replaced nail plate is eventually pushed out as a new nail grows. It is believed that the replaced nail plate acts as a splint to hold open the nail fold and protect the repair. However, a recent retrospective study of nail bed repairs in children reported a higher infection rate in the nail replaced (7.8%, 4 of 51) versus nail discarded groups (0%, 0 of 60).[2] There were also significantly more hospital visits and a longer overall follow-up period needed in the nail replaced group compared with the nail discarded group. The hypothesis is that the replaced nail plate acts as a foreign body, which increases the infection risk and wound problems.

A recent Cochrane review found no randomised trials and concluded there was a lack of evidence to inform all key treatment decisions in the management of fingertip entrapment injuries in children.[3] Our patient/parent survey identified normal nail regrowth and long-term cosmetic appearance, along with infection risk as the most common concerns following surgery.[1] In

2015, we performed a pilot study (Nail bed INjury Assessment Pilot (NINJA-P)) to inform the design and conduct of a definitive trial comparing replacing or discarding the nail after nail bed repair.[4] NINJA-P recruited 60 participants (age range <1–16 years) at four hand surgery centres over 4 months. Participants completed follow-up to 4 months. This successful pilot enabled us to demonstrate the viability of a large randomised trial in an area where such trials are rare. It has also enabled us to refine the main trial design including optimising timing and mode of follow-up as well as providing data, which informed the sample size calculation.

The Nail bed INjury Analysis (NINJA) trial seeks to answer the question 'should the nail plate be replaced intraoperatively or discarded after nail bed repair in children, as evaluated by surgical site infections and appearance of the nail (coprimary outcome measures)?' This will help determine whether the simple act of discarding the nail improves the appearance, reduces infection rates and reduces hospital attendances for thousands of children undergoing this operation every year.

### Good clinical practice
The NINJA randomised controlled trial (RCT) will be carried out in accordance with Medical Research Council Good Clinical Practice and applicable UK legislation while following the protocol V.3.0 (4 June 2019).

### Consolidated standards of reporting trials
The trial will be reported in line with the Consolidated Standards of Reporting Trials statement using the non-pharmacological treatment interventions extension.

### Objectives
#### Primary objectives
To assess the effects of replacing or discarding the fingernail in children undergoing surgical nail bed repair by comparing the risk of early nail-related surgical site

**Table 1** Objectives and outcome measures

| Objectives | Outcome measures | Time point(s) of evaluation of this outcome measure (if applicable) |
| --- | --- | --- |
| *Primary objectives*<br>To assess the effects of replacing or discarding the fingernail by comparing the risk of infection and cosmetic appearance. | Incidence of surgical site infection (clinical assessment around 7 days and participant or parent/guardian reported with clinical notes at a minimum of 4 months if information is relevant to earlier time period). | 7 days |
| | Oxford Finger Nail Appearance Score assessing nail appearance at a minimum of 4 months, considering five domains (shape, adherence, eponychium, surface quality and presence of split). | At a minimum of 4 months |
| *Secondary objectives*<br>To assess whether there is a difference in participant/parent and guardian reported health-related quality of life according to whether the nail is replaced or discarded. | EuroQol EQ-5D-Y and proxy completed by the child/parent or guardian according to the age of the participant. | Baseline, 7 days and a minimum of 4 months |
| To assess whether there is a difference in participant or parent/guardian-reported pain experienced between replacing and discarding the nail. | The level of pain experienced by the child at their first dressing change assessed by the child or parent/guardian (3-point Likert scale for children). | 7 days |
| To conduct a parallel within-trial economic analysis to assess the cost-effectiveness (including resource use) of replacing versus discarding the nail. | Healthcare resource use such as increased hospital visits, dressing and antibiotic use and in some cases hospital readmission and repeat surgery. | 7 days and 4 months |
| To assess if any surgical site infection has occurred within the 4 months since surgery. | Participant or parent/guardian-reported incidence of infection with clinical notes confirmation. | At a minimum of 4 months |
| To assess participant/parent satisfaction with nail healing | Child or parent/guardian satisfaction with nail healing (3-point Likert scale for the children and a VAS score for the parents/guardians). | At a minimum of 4 months |

VAS, Visual Analogue Scale.

1. Eligible participants identified
2. Participants provided with information & recruited
   a. Patient Information Sheet (age appropriate-language)
   b. Parent/guardian Information Sheet
3. Participant consented
   a. Consent (for parents/guardians)
   b. Assent (for the child)

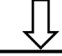

4. Participant randomised
   a. Patient details and randomisation form (hospital)
5. Pre-surgery data collection
   a. Health questionnaire (hospital)

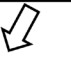 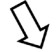

6. Nail bed surgery with nail _replaced_
   a. operative form

6. Nail bed surgery with nail _discarded_
   a. operative form

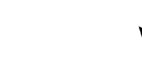

7. Post-operative 7 day visit for dressing change
   a. Follow Up CRF (clinical visit)
   b. Retrospective Baseline QoL questionnaire (clinical visit)

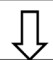

8. Follow up minimum of 4 months post-op
   a. Follow Up questionnaire (emailed/posted to parent/guardian)
   b. Health Resource questionnaire (emailed/posted to parent/guardian)
   c. Photograph of affected nail and healthy nail submission (via online portal)

**Figure 1** Trial flowchart. CRF, case report form; QoL, quality of life.

infection and cosmetic appearance at a minimum of 4 months (table 1).

## Secondary objectives

a. To assess whether there is a difference in participant/parent/guardian-reported health-related quality of life according to whether the nail is replaced or discarded.

b. To assess whether there is a difference in participant/parent/guardian-reported pain experienced between replacing and discarding the nail at first dressing change.

c. To conduct a parallel within-trial economic analysis to assess the cost-effectiveness of replacing versus discarding the nail.

d. To assess if any late nail-related surgical site infection (eg, osteomyelitis) has occurred within the last 4 months (in addition to early infection with the first 7 days).

e. To assess participant/parent/guardian satisfaction with nail healing at a minimum of 4 months.

## METHODOLOGY

NINJA is a multicentre, pragmatic two-arm parallel group superiority RCT. A minimum of 416 patients will be recruited from up to 30 National Health Service (NHS) hand surgery units across the UK over 18 months (July 2018–December 2019). Participants will be randomised to either have the nail plate replaced or discarded following repair of the nail bed injury. They will be followed by their local clinics at the first routine clinic appointment (around 7 days postoperation) and will report additional treatments received in the following 4 months. Participants and parents (or guardian) will complete questionnaires at the clinic appointment and report (parent or guardian) questionnaires at around 7 days and at 4 months via electronic post. Parents will provide photos of the injured and matched opposite finger at a minimum of 4 months following surgery. If problems are reported via the parent questionnaire, the clinics will be queried for need of reporting of additional treatment. The trial will run for 3 years. A flowchart depicting the trial process is shown in figure 1.

### Outcome measures
#### Primary outcome measures
##### Surgical site infection at 7 days

The presence of a surgical site infection (SSI) at 7 days postsurgery will be collected. The principal means of data collection will be via a clinical research nurse or attending surgeon assessment of the child's fingertip for absence or presence of infection at the surgical site at the clinical visit. It is often difficult to accurately assess infection in very young children and as this is a pragmatic study clinical judgement of infection will be used and is likely to be based on redness, localised pain, presence of pus and fever. Treatment with antibiotics and return to theatre for infective complications will also suggest a diagnosis of infection. Simple inflammation and non-specific pain following this trauma and surgery are not always markers for surgical site infection in this patient population. Where appropriate, other data sources (eg, 4 months parent/guardian questionnaire) will be used to supplement this for occurrence of an SSI within the relevant timeframe.

#### Cosmetic appearance of the nail

The cosmetic appearance of the fingernails will be assessed using the Oxford Finger Nail Appearance Score at least 4 months postrandomisation. The score will be a sum of the five components, nail shape, nail adherence, eponychium, nail surface and nail plate split. Each component will be given a score of one if it is deemed to be same as the opposite finger or not having the defect and a score of 0 if the fingernail is deemed to be worse than the opposite finger or if the defect is present. The best total score will be a 5, and the worst possible score is 0.

The assessors will be made up of surgical trainees, specialist registrars and hand physiotherapists, who will review the photographs submitted at the minimum 4 months follow-up time point. The assessors will be blinded to the intervention the participant received, although they may have been involved in the trial at a participating site (ie, recruitment, surgery or follow-up). Assessors will complete training on the Oxford Finger Nail Appearance Score. The first batch of approximately 50 photographs will be assessed for quality control purposes, and if needed, modification to assessment training, instruction to parents and the Oxford Finger Nail Appearance Score may be necessary. If so, the first batch of photographs will be reassessed to the new standards. The appearance of the nail will be assessed on the case report form (CRF) using the Oxford Finger Nail Appearance Score, and the development of this was informed by the Zook Nail Classification Scale.[5]

### Secondary outcome measures
#### Health-related quality of life
The EQ-5D-Y is a validated, child-friendly, health-related quality of life questionnaire consisting of five domains related to daily activities with three-level answer possibilities. This will be completed by the patient (or via parent/guardian proxy depending on the child's age) at baseline, 7 days and 4 months postrandomisation.

#### Pain at dressing change
The level of pain experienced by the child at their first dressing change which occurs at 7 days will be assessed using a 3-point Pain Likert scale for children (based on the Wong Baker Scale). This will be completed by the patient or a parent/guardian proxy.

#### Cost-effectiveness
A health resource use questionnaire will be completed by the parent/guardian at 7 days and 4 months postrandomisation. This will collect information on hospital visits, dressing and antibiotic use and hospital readmission and repeat surgery.

#### Surgical site infection by 4 months
The presence of an SSI during the 4-month period postrandomisation will be assessed. In addition to the clinical assessment at 7 days, the patient's parent/guardian will be asked if the patient experienced any problems postsurgery. This will then be referred back to sites, where appropriate, to obtain confirmation from clinical notes and, if necessary, general practitioner notes. This will capture any surgical site infections which occur after the usual expected timeframe in which infections would normally present.

#### Participant/parent satisfaction with nail healing
A patient assessment of the nail appearance (3-point Likert scale for children) will be used to measure patient-reported satisfaction with the healing of the nail at 4 months postrandomisation. If the child cannot complete this score, a Visual Analogue Scale in the form of a measured line with a continuous scale (from 0 to 100) anchored by two verbal descriptors for each extreme symptom will be used as a patient proxy for measuring satisfaction with nail healing.

### Study population
#### Inclusion criteria
► Male or female, aged below 16 years old at the time of presentation to the participating hospital.
► Nail bed injury occurring within 48 hours of presentation at a trial centre believed to require surgical repair by the surgical team. This includes sharp lacerations, stellate lacerations, crush and avulsion injuries of the nail bed, injuries involving the sterile and/or germinal matrix, nail bed injuries with an associated pulp laceration and/or with an associated 'tuft' fracture of the distal phalanx.
► Patients whose parent or legal guardian consent to their inclusion in the trial and are willing to complete follow-up, including photographs.
► Sufficient understanding of the child and parent/guardian participant information sheets as deemed by recruiting team at local sites.
► Single-digit nail bed injury.

#### Exclusion criteria
The participant will not enter the trial if any of the following apply.
► Patients present with an infected nail bed injury.
► Patients have an underlying nail disease or deformity in the injured or contralateral finger prior to the injury.
► Patients have an associated distal phalanx fracture, requiring fixation with a Kirschner wire.
► Patients with an amputation of the distal fingertip including all or part of the nail bed.
► Patients with loss of part or all of the nail bed, requiring a nail bed graft or flap reconstruction.
► Previous NINJA trial participants.
► Patients with nail bed injuries to more than one digit.

### Recruitment and consent
Trial participants will be prospectively recruited from the participating hospitals. Initial assessment will take place in the accident and emergency/minor injuries department or paediatric ward. The clinical team will identify any potential participants and refer on to the research team for further information. The research team will obtain informed consent. Screening logs will be maintained at each site. Reasons for non-participation and/or ineligibility will be documented.

Parents/guardians will be given an information sheet and have the trial explained to them by the researcher. Children will also be provided with age appropriate information to include them in the consent process. Consent for medical photography will be included as part of the consent process for the research team to analyse

participant submitted photographs, and agreement to return follow-up questionnaires and submit a photograph at a minimum of 4 months postsurgery will be part of the inclusion criteria.

## Data collection

The baseline assessment will be on the day of the operation, before randomisation but after consent to participation. Participant demographics will be recorded on the CRF by the assessing surgeon either in the emergency department or on admission to the paediatric ward. Follow-up assessments will involve a clinical appointment around 7 days postoperation and a participant reported questionnaire, sent via text, email or post, at around 7 days postoperation and 4 months (table 1 and figure 1).

## Randomisation and blinding

A web-based randomisation system will be provided by the Oxford Clinical Trials Research Unit (OCTRU). The allocations will be computer generated with a 1:1 ratio and stratified by site using random permuted blocks of varying size within stratum. Randomisation will take place when the participant is in the anaesthetic room just prior to surgery or as close to the surgery time as possible by a good clinical practice (GCP) trained member of the team.

This is an open trial, since those delivering the care will not be blinded to the intervention the participant has been allocated to. This is because a replaced nail can take several weeks to loosen and fall off once a new nail has grown out, and therefore the treatment received will be obvious within this timeframe. Therefore, the assessment of the photographs for cosmetic appearance at a minimum of 4 months will be done by independent assessors who can at that time point be blinded.

## Operative assessment

At the time of surgery, the operating surgeon will classify the nail bed injury according to the system used and tested in the pilot.[4]

## Interventions

### Nail bed repair

In both groups (nail plate replaced or nail plate discarded), the nail bed repair will be performed using 6/0 or 7/0 interrupted Vicryl Rapide (Johnson and Johnson Medical, Livingston, West Lothian, UK) or equivalent sutures. This is a pragmatic trial. The following decisions will be left to the discretion of the surgical team responsible for the participant but recorded on the CRF:

► The type of anaesthetic used (general anaesthetic, local anaesthetic or both).
► Perioperative antibiotics given, if any.
► Type and duration of tourniquet used.
► Type of surgical preparation solution and wash used.
► Type of dressing applied. In practice, this is usually a combination of a non-adherent dressing, absorbent layer and a top layer of fabric-based dressing to keep the digit covered.

If the surgeon has to perform a procedure(s), which was part of the exclusion criteria, this will be recorded on the CRF. This is an extremely unusual event as the vast majority of these procedures (eg, fracture fixation with a Kirschner wire, need for a composite graft or nail bed graft) are predictable preoperatively. These participants will be analysed within the intention to treat analysis of the trial. In both groups, the fingertip will be dressed with a non-adherent dressing. The operating surgeon will add to the CRF the following data: the type of nail bed injury, whether the nail plate was replaced or discarded, whether a nail substitute was used, what, if any, antibiotics were given perioperatively and what postoperative antibiotic regime is planned.

### Nail plate replaced

In the nail plate replaced group, the nail plate will be secured using a figure-of-eight vicryl rapide suture. If the nail plate cannot be replaced in a participant randomised to this group, for example if it is too badly damaged, a nail substitute of the operating surgeon's choice will be used and recorded on the CRF.

### Nail plate discarded

In the nail plate discarded group after the nail bed repair, the nail will not be replaced. It will be discarded appropriately instead. The washout, debridement and suturing procedures will be the same as described for the first group.

## Safety reporting

Data on adverse and serious adverse events will be recorded and their severity and frequency will be assessed. Standard Health Research Authority (HRA) safety reporting measures will be adhered to. The OCTRU conducted a risk assessment prior to the trial starting. Issues raised have been addressed within the current approved protocol, and procedures have been planned to monitor the ongoing risks of the trial. A risk proportionate approach will be used within this trial. Central monitoring of trial procedures will be embedded into the trial conduct and management, including instituting a trial steering committee (TSC) and data monitoring committee (DMC). The TSC and DMC will agree their respective terms of reference. No formal statistical interim safety analysis has been planned for in the design or are anticipated given the nature of the trial. The trial may be monitored or audited in accordance with the current approved protocol, GCP, relevant regulations and standard operating procedures. The trial will be subject to audit according to OCTRU's audit programme.

## End of trial

The end of trial is the date of the last follow-up of the last participant.

## Analysis

### Statistical analysis

Principal analyses will be on an 'as randomised' basis retaining participants in their randomised allocation groups irrespective of compliance to the allocation. A two-sided 5% significance level will be adopted with associated 95% CIs whenever possible using appropriate summary measures (eg, number of events and percentage for binary measures). The principal analyses will also be carried out on a complete case basis with sensitivity to missing data explored for the primary outcomes.

### The number of participants

Sample size calculations are based on the coprimary outcomes of SSI and cosmetic appearance at a minimum of 4 months, measured via the Oxford Finger Nail Appearance Score, the development of which was informed by the Zook Nail Classification Scale[5]—a 0–5 ordinal summary score reflecting optimal or suboptimal appearance across the five classification domains. Pilot data from our NINJA-P trial[4] showed a substantial proportion of participants did not have nails with optimal appearance (approximately 35% had two or more suboptimal aspects of appearance, ie, score of three or less). Based upon a clinically relevant difference of 15% more achieving the optimal appearance score of 5 (from 35% to 50% with a corresponding shift in the other score values) and using a two-sided significance level of 0.05, 332 (166 per group) are required to obtain 90% power based on a Mann-Whitney U test. After allowance for 20% missing data, a total of 416 participants (208 in each group) are required. This calculation was carried out using an extended version of the Excel spreadsheet provided by Walters[6] to allow for a 6-point ordinal outcome. Based on a lower overall level and a smaller difference in the proportion with an SSI than the one observed in the Miranda[2] observational study (8% vs 1%), this sample size is also sufficient for 90% power at the two-sided 5% significance level. This latter calculation was carried out in Stata V.14 using the power two prop command.

### Analysis of outcome measures

As multiple assessors will be reviewing each photograph using the Oxford Finger Nail Appearance Score, the median of the assessors' total scores will be used as the rating for each photo to account for any variability in scores. These will then be analysed using a Mann-Whitney U test (with a 95% CI for the median also calculated). A secondary more complex ordinal regression model will also be used to estimate the difference across the ordinal scale and allow subgroup analyses.

SSI will be compared using logistic regression adjusted for site. If the number of events is too low for adjustment, univariate logistic regression will be carried out. Prespecified subgroup analysis will be carried out according to preoperative antibiotic use using a treatment-by-subgroup interaction extending the aforementioned regression models for the coprimary outcomes. Secondary outcomes will be analysed using generalised linear models as appropriate. Further details of the planned statistical analyses will be specified in a Statistical Analysis Plan, which will be finalised prior to the unblinding of data to NINJA investigators. Available data will be used up to the point of withdrawal whenever possible.

### Economic analysis

A within-trial cost-utility analysis comparing nail replacement with nail discarding will be conducted from the UK NHS and Personal Social Services perspective in the base case (or primary) analysis.[7]

Resource use for the surgery will be recorded by the research team in the CRF while data for the economic evaluation will be collected from the trial questionnaires given to participants at around 7 days and at a minimum of 4 months after randomisation. Unit cost of this resource use will be sourced from the latest NHS Supply Chain Catalogue, NHS Reference Cost and British National Formulary. Where appropriate, the cost of health resource use per patient will be computed by multiplying the frequency of health resource use with the unit cost of each resource item.

Health-related quality of life will be estimated using the EQ-5D-Y questionnaire at baseline, at around 7 days and at a minimum of 4 months. The EQ-5D-Y user guide instructions will be followed so that children are given age-appropriate questionnaires to answer.[8]

A cost-utility analysis (excluding the participants below the age of 2) will present outputs of the analyses in terms of incremental cost-effectiveness ratio (ICER) where the National Institute for Health and Care Excellence cost-effectiveness threshold of £20 000–£30 000 per additional quality-adjusted life year (QALY) will be applied. Given the methodological limitations surrounding preference-based outcomes measurement in young children, a cost-effectiveness analysis will also be conducted (for the entire sample) where outputs will be expressed in terms of incremental cost per surgical site infections prevented.

If data are missing at random, multiple imputation analysis will be performed to avoid bias associated with the complete case analysis. We assume no outcome differences in terms of QoL, pain and complications beyond the trial period, therefore no longer time perspective will be considered.

Sensitivity analysis such as extending the study perspective to societal perspective and assessing the impact of missing data on the ICERs will be performed. To assess sampling uncertainty on the ICERs and varying willingness-to-pay levels for an additional QALY, probabilistic sensitivity analysis (PSA) will be performed. Results from the PSA will be presented in cost-effectiveness acceptability curves, which will be generated via non-parametric bootstrapping.

### Patient and public involvement

To inform study design, 30 parents of children with nail bed injuries were surveyed. The survey identified normal

regrowth of the nail, infection and long-term appearance as the most common parental concerns following nail bed surgery.[1] Subsequently, a focus group and youth group refined follow-up methods, types of study material, as well as which outcomes were important. To ensure ongoing patient and public involvement, a patient/carer representative is actively involved in general trial management. In addition, further independent patient/carer representatives are members of the steering committee.

## Ethics and dissemination

The participants in this trial are children and consent for them to take part will need to be obtained from their parent or legal guardian by a GCP trained research team member. If a child wishes not to take part in the trial, this will be respected. Personal information will be handled confidentially in line with European Union General Data Protection Regulation (GDPR) regulations. Any publication arising out of the trial will follow the National Institute for Health Research publication policy.

**Author affiliations**
[1]Department of Plastic Surgery, Imperial College Healthcare NHS Trust, London, United Kingdom
[2]Nuffield Department of Orthopaedics, Rheumatology and Musculoskeletal Sciences, University of Oxford, Oxford, United Kingdom
[3]Department of Plastic Surgery, Frimley Health NHS Foundation Trust, Slough, United Kingdom
[4]St Andrew's Centre for Plastic Surgery and Burns, Mid Essex Hospital Services NHS Trust, Chelmsford, United Kingdom
[5]Department of Plastic and Reconstructive Surgery, Guy's and St Thomas' NHS Foundation Trust, London, United Kingdom

**Contributors** AJ: developed research question, contributed to protocol design, first draft of manuscript. AJo refined the protocol and edited the manuscript. CC contributed to the protocol design. AS and AVHG developed the original research question and protocol and edited the manuscript. MDG contributed to research question, protocol design and edited the manuscript. MD: health economic evaluation. MEP: health economic evaluation. JRS and BS: sample size and statistical analysis. JC: sample size and statistical analysis. DB: developed the research question. All authors reviewed and agreed the final manuscript.

**Funding** AJ, MDG, CC, AS, JC, DB, AVHG obtained grant funding for this project. This project was funded by the UK National Institute for Health Research (NIHR) Research for Patient Benefit (RfPB) grant (PB-PG-1215-20041) and was supported by the NIHR Biomedical Research Centre.

**Competing interests** None declared.

**Patient consent for publication** Not required.

**Ethics approval** This trial is conducted in accordance with the principles of the Declaration of Helsinki, with relevant regulations and with Good Clinical Practice. It has been approved by the South Central Research Ethics Committee (Berkshire-B, 04/06/2019, ref: 18/SC/0024).

**Provenance and peer review** Not commissioned; externally peer reviewed.

**ORCID iDs**
Matthew D Gardiner http://orcid.org/0000-0002-8058-4186
Melina Dritsaki https://orcid.org/0000-0002-1673-3036
Jamie R Stokes http://orcid.org/0000-0002-5279-2332
Jonathan Cook https://orcid.org/0000-0002-4156-6989

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
