## [Reviewer comments · BMJ Open]

ARTICLE DETAILS

TITLE (PROVISIONAL)	The NINJA Trial - should the nail plate be replaced or discarded after nail bed repair in children? Protocol for a multi-centre randomised control trial
AUTHORS	Jain, Abhilash; Jones, Amy; Gardiner, Matthew; Cooper, Cushlar; Sierakowski, Adam; Dritsaki, Melina; Png, May ee; Stokes, Jamie; Shirkey, Beverly; Cook, Jonathan; Beard, David; Greig, Aina

VERSION 1 – REVIEW

REVIEWER	Julie Bruce Warwick Clinical Trials Unit, University of Warwick, UK
REVIEW RETURNED	29-May-2019

GENERAL COMMENTS	This is an interesting protocol on an important topic given the volume of nail bed injuries in children. Good to hear about the reconstructive surgery trial network which will support recruitment to clinical trials. The authors should be congratulated on setting up the first UK trial involving this network. I have a number of comments but these are mostly clarifications thus the recommendation of minor rather than major revision. There are a number of inconsistencies and some more explanation is required. These comments are made in the spirit of improving the protocol to ensure that when it comes to publishing, the final reporting process will be easier. I realise the trial is probably well into recruitment by now but the protocol needs to be more explicit, particularly with regards to the primary outcome rather than relying upon readers to return to the pilot study (which i have not done). Good luck with the trial. NINJA protocol: Differences between submitted protocol and ISRCTN registration • Protocol is missing the dates of the study recruitment, states on ISRCTN that started in 2017 and will complete in April 2020• Sample size differs between registry (n=464) and protocol (N=416)• Registry states PEDs Quality of life scale will be used in conjunction with EQ5D-Y but not in protocol Abstract page 5 • Line 6 – suggest replace “we will look at’ – with we will investigate / this study will investigate• Line 22 – add in surgical repair to the aim, e.g. whether replaced or discarded after surgical nail bed repair• Line 36-39 add timing of primary outcome.• Line 39, replace ‘level of pain’ with pain intensity
--

	 • Line 43 states minimum of 416 patients yet this doesn't match with trial registry site • Line 46 – minor point but does imply that the parents/carers have undergone an operation • Line 48 – assume this is SSI not just infection (ie. Other infective complications) • Line 50 – states will be asked to “send in photos of their fingernails” but doesn't explain what these will be used for. Is this for independent assessment of wound healing, cosmetic appearance etc?? Page 6, line 12/13 – inform whether the practice of intraoperative surgical replacement of the nail should continue. It was not clear to the reader, at least initially, whether the replacement of the nail was a separate surgical procedure undertaken a few days after surgery (2 stage process). Introduction  • Page 7 Line 12 – rephrase ‘the suggestion’. Could you perhaps specific as a hypothesis? • Page 7, line 52 – add in age range for pilot study • Page 8, lines 8-12 – should the nail plate be replaced intraoperatively(?) or discarded after surgical nail bed repair.... • Page 8, Line 12 – it is not overall infective complications, it is surgical site infection, unless you are picking up respiratory, blood infections etc? • States on page 8 that the protocol is version 1 dated 2016? On page 6 states that the protocol version 1 was dated May 2019. Needs correction.  • Page 8, lines 51 – add in surgical nail bed repair and replace infection with nail-related surgical site infection. Suggest you be consistent with terminology for SSI throughout, the focus of the study is nail bed infection not respiratory / blood etc.  • Page 9 – line 3, confused by the participant/parent and guardian statement. Is this correct? Should this be participant or the parent/guardian? Assume you only want one not both?  • Page 9, line 22, add in nail-related SSI. Confused by this being added as a secondary as you have already specified “infection and cosmetic appearance at 4 months” in the primary objectives on previous page? But here it implies that early SSI, within 7-10 is the primary. Suggest clarify the primary objective on page 8, lines 53.  • Page 9, Line 26/27 – no mention of guardian here but is listed for previous objectives. Methods Page 9, line 37 – 416 will be recruited Page 9, line 58 – states providing photos but unclear here whether this is the parents providing photos at 7 days or medical illustration. Outcome measures Primary: SSI at 7-10 days The lack of information about HOW SSI will be determined is a major omission given this is the primary outcome. Superficial SSI as defined by CDC clearly gives criteria for determining infection (e.g. specific combinations of redness, inflammation, pain, tenderness, pus, fever etc). How does this apply to nail bed injury? What definition is being used? No reference given. The nurse or surgeon will make the judgement of SSI at 7-10 days, but are all the criteria being collected individually then an algorithm applied? Sorry to be pedantic here but this is the primary outcome & you have got a
--	---

	reviewer who is aware of the challenges of capturing wound-related outcomes/SSIs in clinical trials. You may need some checks of agreement – you are doing that for the other co-primary but not for SSI. The CDC definition for superficial SSI is that occurring within 30 days. Cosmetic appearance Page 20, lines 34-37. No explanation of modified version, how/why was modified or reference (although a ref is given later). Suggest restrict panel of assessors to two or three people.  • Page 11, lines 27-28. EQ-5D-Youth measure. The trial registry states that the PEDs Quality of life scale will be used in conjunction with EQ5D-Y but this is not in the protocol. Please amend. • Pain at dressing change – should be first dressing change which assume is in first few days, can you give an indication of roughly when this is expected to be? The Wong Baker FACES scale is either 5 or 10 point scale so would be helpful again to know why or how this was modified to give a 3 point scale – no reference of the version you will be using is given. • Page 11, line 58-59, removed ‘in some cases’ – you will capture this information on everyone? • Page 11, lines 8-9, first batch of 50 photographs. Assume you mean the first batch of 50 participants rather than photos (patients & clinical staff often return multiple images) but this is rather late to amend the protocol. Suggest you do this on a smaller number of 10-20 children. Unclear whether photograph was tested in pilot study. Secondary outcomes  • Page 12, line 6. SSI by 4 months, assume this is any SSI occurring over the 4 months rather than SSI at the 4 months point? Thus rate of infection. Any self-report may need confirmation from GP notes rather than hospital notes. Page 14, lines 17-22. This section is unclear as implied earlier in protocol that parents would take photograph at 7-10 days. Assume consent for medical photography is for clinical team to photograph at clinic but then parental photograph at 4 months. Needs some clarification. Page 14, lines 29-31. Explanation in text does not match flow chart where shows that patient is randomised and then completes baseline information. The flow chart needs correcting. Page 14, lines 35/6. “sees the patient” – rephrase as surgical review Page 15, lines 17/22. Explains blinding & rationale for open trial, but who will assess photographs that are taken at 7-10 days and is there any need to take a photo at this time point? A bit confusing as perhaps there is no early photography. Page 16: Nail bed replaced. Any brief info in this section or above on usual care for keeping dressings on? Similarly for nail bed discarded, is it usual practice to cover or leave open. Realise is pragmatic but brief info on usual care would be helpful. Page 17: lines 3-4. “Data on complications will be recorded” – do you mean adverse events and serious adverse events here? Please use standard CTU terminology.
--	---

	Page 18. The number of participants (sample size?) Page 18, lines 14-16 state that cosmetic appearance is using a modified “Oxford cosmetic nail score”, first time this outcome has been mentioned, not mentioned in the abstract or outcomes /data collection section? Sample size based on 15% difference in optimal cosmetic appearance. Lines 44/45- SSI is based on Miranda study which had 4 infections out of 111 children (3.6%) thus implies you expect to find 15 infections out of 416 children. But I’m not too clear what the difference is, is it 7% difference in SSI between treatment arms? How are you handling the co-primary outcomes e.g. the SSI is measured at 7 days and you will have more data at that time point than from photos returned at 4 months postop. How do you determine effectiveness if findings are opposite direction, (more early infection but look better overall – assume the QoL will dominate). This is not covered at all – more of an issue too because of the differences in timing & potential for missingness. Analysis of outcome measures Page 19 line 21. Unlikely that you can account for all of these factors when you expect to only have up to approx. 15 SSIs overall. Page 20, line 19-20. The user guide instructions will be followed “as much as possible” – either they will be followed or they wont. Figure – flow chart You give age strata in the flow chart but this is not mentioned anywhere in the protocol.  • Randomisation is intraoperatively? • Confusing about baseline data collection – states here completed in clinic but after randomisation which is in theatre? • Why is clinical procedure added to the intervention boxes – this is intraoperative so suggest add this or add in surgery somewhere.
--	--

REVIEWER	Daniel M Weber Pediatric Hand Surgery University Children's Hospital Zurich Switzerland
REVIEW RETURNED	12-Jun-2019

GENERAL COMMENTS	The authors have submitted an excellent study protocol regarding a clinically relevant subject, clear primary and secondary objectives and a prospective randomized study design that is likely to answer the study questions. A strength of the design is that a pilot study with 60 patients was performed. A weakness of the design is the fact, that the results at the end of the study will be based on photographs by parents or patients without a clinical follow-up. The quality of some photographs might be inappropriate to allow an analysis with the Zook score. The high number of centers involved (30) may be a challenge.
--

VERSION 1 – AUTHOR RESPONSE

We thank the reviewers for their very helpful comments. They have led to an improvement manuscript and a clearer and fairer report of the study protocol.

NINJA protocol: AMY can you please address?

Differences between submitted protocol and ISRCTN registration

- Protocol is missing the dates of the study recruitment, states on ISRCTN that started in 2017 and will complete in April 2020. We have added text to the protocol to state the original recruitment period. We have updated the registry given very recent decision to extend recruitment and therefore the end of the study.
- Sample size differs between registry (n=464) and protocol (N=416). We thank the reviewer for pointing this out and apologise for this very unfortunate typographical error. The target sample size has always been 416 since the study was funded. This error creep into the registry submission, it is not clear how. All protocol versions and the ethics submission have always had n=416 for this target sample size.
- Registry states PEDs Quality of life scale will be used in conjunction with EQ5D-Y but not in protocol We had initial collected PedsQL scale for a subset of participants. However, for the response explained below we have changed this. While we will have some data on PedsQL, it will not form the basis of the analysis.

Abstract page 5

- Line 6 – suggest replace “we will look at’ – with we will investigate / this study will investigate done
- Line 22 – add in surgical repair to the aim, e.g. whether replaced or discarded after surgical nail bed repair done
- Line 36-39 add timing of primary outcome. done
- Line 39, replace ‘level of pain’ with pain intensity done
- Line 43 states minimum of 416 patients yet this doesn’t match with trial registry site Please see our response above on this topic.
- Line 46 – minor point but does imply that the parents/carers have undergone an operation “or depending on age their” added to clarify
- Line 48 – assume this is SSI not just infection (ie. Other infective complications) “surgical site” added to clarify
- Line 50 – states will be asked to “send in photos of their fingernails” but doesn’t explain what these will be used for. Is this for independent assessment of wound healing, cosmetic appearance etc?? “to assess cosmetic appearance” added

Page 6, line 12/13 – inform whether the practice of intraoperative surgical replacement of the nail should continue.

It was not clear to the reader, at least initially, whether the replacement of the nail was a separate surgical procedure undertaken a few days after surgery (2 stage process). “at the time of nail bed repair” added

Introduction

- Page 7 Line 12 – rephrase ‘the suggestion’. Could you perhaps specific as a hypothesis? Changed to “hypothesis”
- Page 7, line 52 – add in age range for pilot study added
- Page 8, lines 8-12 – should the nail plate be replaced intraoperatively(?) or discarded after surgical nail bed repair.... added
- Page 8, Line 12 – it is not overall infective complications, it is surgical site infection, unless you are picking up respiratory, blood infections etc? Changed a suggested to “surgical site infections”

- States on page 8 that the protocol is version 1 dated 2016? On page 6 states that the protocol version 1 was dated May 2019. Needs correction. Done
- Page 8, lines 51 – add in surgical nail bed repair and replace infection with nail-related surgical site infection. Suggest you be consistent with terminology for SSI throughout, the focus of the study is nail bed infection not respiratory / blood etc. changed as suggested
- Page 9 – line 3, confused by the participant/parent and guardian statement. Is this correct? Should this be participant or the parent/guardian? Assume you only want one not both? Amended to clarify we only want one.
- Page 9, line 22, add in nail-related SSI. Confused by this being added as a secondary as you have already specified “infection and cosmetic appearance at 4 months” in the primary objectives on previous page? But here it implies that early SSI, within 7-10 is the primary. Suggest clarify the primary objective on page 8, lines 53. Added “late nail-related surgical site infection (e.g. osteomyelitis)” to clarify
- Page 9, Line 26/27 – no mention of guardian here but is listed for previous objectives. Added

Methods

Page 9, line 37 – 416 will be recruited. We are unclear what the point being raised here. If related to apparent disagreement we have addressed above.

Page 9, line 58 – states providing photos but unclear here whether this is the parents providing photos at 7 days or medical illustration. Sentence to clarify added “Parents will provide photos of the injured and matched opposite finger at a minimum of 4 months following surgery.”

Outcome measures

Primary: SSI at 7-10 days

The lack of information about HOW SSI will be determined is a major omission given this is the primary outcome. Superficial SSI as defined by CDC clearly gives criteria for determining infection (e.g. specific combinations of redness, inflammation, pain, tenderness, pus, fever etc). How does this apply to nail bed injury? What definition is being used? No reference given. The nurse or surgeon will make the judgement of SSI at 7-10 days, but are all the criteria being collected individually then an algorithm applied? Sorry to be pedantic here but this is the primary outcome & you have got a reviewer who is aware of the challenges of capturing wound-related outcomes/SSIs in clinical trials. You may need some checks of agreement – you are doing that for the other co-primary but not for SSI. The CDC definition for superficial SSI is that occurring within 30 days.

We recognize the importance of the CDC definition of infection and its use in diagnosis of infection and strict adherence to this was considered. However, it is often very difficult in clinical practice to apply this to very young children who have had surgery for trauma, especially those under 2 years old. Having debated this with many clinicians a pragmatic approach was adopted and a decision was made to rely on the assessing clinicians judgment of infection rather than defined definitions. For example, a 12 month old child cannot vocalize the site of pain and may present with a mild fever from an upper respiratory tract infection, yet on dressing change have erythema secondary to the original trauma deemed to be within the boundary of what would be acceptable 1 week following surgery. Vague pain, fever and redness could in this case suggest infection but clinicians may feel that these symptoms are not indicative of an SSI and choose not to treat as a nail bed related infection. We have therefore added the following section to the paper to clarify this

“It is often difficult to accurately assess infection in very young children and as this is a pragmatic study clinical judgment of infection will be used and is likely to be based on redness, localized pain, presence of pus and fever. Treatment with antibiotics and return to theatre for infective complications

will also suggest a diagnosis of infection. Simple inflammation and non-specific pain following this trauma and surgery are not always markers for surgical site infection in this patient population.”

Cosmetic appearance

Page 20, lines 34-37. No explanation of modified version, how/why was modified or reference (although a ref is given later). Suggest restrict panel of assessors to two or three people.

We used a score which was developed based upon the Zook (Zook et al ref no. 4 in the paper) the preparatory NINJA study. However, it was showed to have poor agreement between assessors when making assessments of the cosmetic appearance of the nail in this study. Given this we sought to produce a simplified score and all revised the process for collecting the photos, and this has occurred in two stages – first prior to setting up of the main NINJA trial, and two, partway through – the latter in particular has focused upon how to train the assessors and achieve consistency in assessment. The original Zook based score had higher dimensionality, with two of the components (shape and surface) having five possible scores, and the other three components having 4, 3 and 2 possible scores (adherence, eponychium and split respectively). The new score has been simplified for the full trial by adjusting each component to have only two possible scores each. Shape, surface and eponychium will now be scored as either “Identical to opposite” or “Not identical to opposite” (in comparison to the same nail on the opposite hand); adherence will be scored as either “Complete” or “Incomplete”, and split will be scored as either “Present” or “Absent”. Assessors will also be provided with training on how to use the score, as well as a reference sheet to aid them in completing their assessments. Five assessors will score each photo, as an odd number of scores is required in order to take the median as the overall score for the photo. Five assessors are being used instead of three to decrease the possibility of an erroneous result. Given these very substantial changes to the original Zook classification and the changes in the assessment process, on reflection we do not think it is appropriate to continue to describe it even as a “modified Zook score” and therefore have used the name “Oxford Finger Nail Appearance Score” throughout in this revised protocol.

• Page 11, lines 27-28. EQ-5D-Youth measure. The trial registry states that the PEDs Quality of life scale will be used in conjunction with EQ5D-Y but this is not in the protocol. Please amend. PedsQL has been excluded from the latest version of the protocol because the mapping algorithm used to estimate health utilities from PedsQL is age-group specific and there are currently no mapping algorithms available for children aged 7 and under (PedsQL for toddlers and young children). Although EQ-5D-Y has not been validated for children under four, given that there are currently no guidelines to collect HRQoL data from children under four and no valid algorithm to map PedsQL to health utilities, EQ-5D-Y (proxy) was used for children aged 2-4 years. As the amendment was submitted after the study started, a few participants will have completed the PedsQL instead of the EQ-5D-Y (proxy). For these patients, their PedsQL profile will still be reported but their HRQoL will not be included in the cost-utility analysis. This amendment and the reason for excluding PedsQL has also been noted in the health economics analysis plan.

• Pain at dressing change – should be first dressing change which assume is in first few days, can you give an indication of roughly when this is expected to be? Added “which occurs around 7-10 days”

The Wong Baker FACES scale is either 5 or 10 point scale so would be helpful again to know why or how this was modified to give a 3 point scale – no reference of the version you will be using is given. We previously used the Wong-Baker scale in the pilot study which preceded this main trial. However, based upon the experience of that study and feedback from the parent/guardian/patient group on the

difficult of it use we decided not to use the Wong-Baker but a simplified scale based upon it. We have described it now as “based upon Wong-Baker” which we think is a fairer description given the differences (including having only three levels) as the reviewer points out.

- Page 11, line 58-59, removed ‘in some cases’ – you will capture this information on everyone? done
- Page 11, lines 8-9, first batch of 50 photographs. Assume you mean the first batch of 50 participants rather than photos (patients & clinical staff often return multiple images) but this is rather late to amend the protocol. Suggest you do this on a smaller number of 10-20 children. We mean photos, as participants implies that the first 50 consecutive patients all returned their photographs as expected which is not the case as demonstrated in our pilot. Unclear whether photograph was tested in pilot study. Yes was tested in pilot but modified the photos collected and the assessment process for the main trial.

Secondary outcomes

- Page 12, line 6. SSI by 4 months, assume this is any SSI occurring over the 4 months rather than SSI at the 4 months point? Thus rate of infection. Any self-report may need confirmation from GP notes rather than hospital notes. This has been changed as suggested
- Page 14, lines 17-22. This section is unclear as implied earlier in protocol that parents would take photograph at 7-10 days. Assume consent for medical photography is for clinical team to photograph at clinic but then parental photograph at 4 months. Needs some clarification. Added to clarify: “in order for the research team to analyse participant submitted photographs”

Page 14, lines 29-31. Explanation in text does not match flow chart where shows that patient is randomised and then completes baseline information. The flow chart needs correcting. Done

Page 14, lines 35/6. “sees the patient” – rephrase as surgical review. This has been changed

Page 15, lines 17/22. Explains blinding & rationale for open trial, but who will assess photographs that are taken at 7-10 days and is there any need to take a photo at this time point? A bit confusing as perhaps there is no early photography. There is no photography taken at 7-10 days only at a minimum of 4 months.

Page 16: Nail bed replaced. Any brief info in this section or above on usual care for keeping dressings on? Similarly for nail bed discarded, is it usual practice to cover or leave open. Realise is pragmatic but brief info on usual care would be helpful.

Sentence added “In practice this is usually a combination of a non-adherent dressing, absorbent layer and a top layer of fabric based dressing to keep the digit covered.”

Page 17: lines 3-4. “Data on complications will be recorded” – do you mean adverse events and serious adverse events here? Please use standard CTU terminology. This has been changed as suggested.

Page 18. The number of participants (sample size?) This is documented further down the paragraph “After allowance for 20% missing data, a total of 416 participants (208 in each group) are required.”

Page 18, lines 14-16 state that cosmetic appearance is using a modified “Oxford cosmetic nail score”, first time this outcome has been mentioned, not mentioned in the abstract or outcomes /data collection section? Sample size based on 15% difference in optimal cosmetic appearance. We apologise for the lack of clarity. Please see the response above related to this topic for a fuller

description of our thinking on this topic. In short, we believe it is appropriate to describe this as the “Oxford Finger Nail Appearance Score” and have done so through the revised protocol.

Lines 44/45- SSI is based on Miranda study which had 4 infections out of 111 children (3.6%) thus implies you expect to find 15 infections out of 416 children. But I’m not too clear what the difference is, is it 7% difference in SSI between treatment arms?

How are you handling the co-primary outcomes e.g. the SSI is measured at 7 days and you will have more data at that time point than from photos returned at 4 months postop. Please see the response to the following comment which addresses this point.

How do you determine effectiveness if findings are opposite direction, (more early infection but look better overall – assume the QoL will dominate). This is not covered at all – more of an issue too because of the differences in timing & potential for missingness. We have not adopted any formal decision rules as we feel it is appropriate to assess the outcomes in their own right. The overall conclusion, should the primary outcomes were to disagree in terms of proposed strategy, would have to weigh up the important of the two outcomes and the magnitude of the effect in each. As the reviewer perceptively points out the two primary outcomes are quite different, measured at different times and therefore may have varying levels of missing data. For all of these reasons, it is our view that no simplistic a-priori decision rule would be appropriate and that it is important to evaluate each independently and present the findings in a transparent way.

Analysis of outcome measures

Page 19 line 21. Unlikely that you can account for all of these factors when you expect to only have up to approx. 15 SSIs overall. We thank the reviewer for this helpful point. While there is only factor (site) depending upon how this is dealt with in the model it can be multiple variables. An alternative approach is cluster-robust variance estimate though that too is likely to be susceptible to a low number of events. Accordingly, we have modified the text to clarify that the analysis may have to be univariate: “Surgical site infection will be compared using logistic regression adjusted for site; if the number of events is too low for adjustment univariate logistic regression will be carried-out.”

Page 20, line 19-20. The user guide instructions will be followed “as much as possible” – either they will be followed or they wont. Changed

Figure – flow chart

You give age strata in the flow chart but this is not mentioned anywhere in the protocol.

We apologies for the lack of clarity. There is no Age strata in terms of the randomization. However, some of the consent process information and the outcome data collection process is age-specific (e.g. patient information leaflet and use of EQ-5D-Y). The flow diagram has been revised to improve the clarity.

- Randomisation is intraoperatively?

Randomisation is typically shortly before the beginning of the operation (e.g. in the anaesthetic room.

- Confusing about baseline data collection – states here completed in clinic but after randomisation which is in theatre?

We apologies for the inappropriate and confusing ordering here. Standard clinical baseline information is being collected before randomization (e.g. age, injury details etc). There is some other clinical data which is being collected pre-surgery but may be post randomization related to general medical history and pre-surgery care. However, we are also collecting retrospective “baseline” quality of life assessments at around 7 days. This has been modified to make this clearer.

- Why is clinical procedure added to the intervention boxes – this is intraoperative so suggest add this or add in surgery somewhere.

We apologise for the confusion. Everyone has surgery. We have removed the phrase “clinical procedure” from both boxes as it is clearly not added to the clarity.

VERSION 2 – REVIEW

REVIEWER	Julie Bruce University of Warwick, UK
REVIEW RETURNED	30-Aug-2019
GENERAL COMMENTS	Well done on addressing all comments. Good luck with trial follow-up.